# Asynchronous stochastic convex optimization: the noise is in the noise and SGD don't care

**Sorathan Chaturapruek**[1]     **John C. Duchi**[2]     **Chris Ré**[1]
Departments of [1]Computer Science, [2]Electrical Engineering, and [2]Statistics
Stanford University
Stanford, CA 94305
{sorathan,jduchi,chrismre}@stanford.edu

## Abstract

We show that asymptotically, completely asynchronous stochastic gradient procedures achieve optimal (even to constant factors) convergence rates for the solution of convex optimization problems under nearly the same conditions required for asymptotic optimality of standard stochastic gradient procedures. Roughly, the noise inherent to the stochastic approximation scheme dominates any noise from asynchrony. We also give empirical evidence demonstrating the strong performance of asynchronous, parallel stochastic optimization schemes, demonstrating that the robustness inherent to stochastic approximation problems allows substantially faster parallel and asynchronous solution methods. In short, we show that for many stochastic approximation problems, as Freddie Mercury sings in Queen's *Bohemian Rhapsody*, "Nothing really matters."

## 1   Introduction

We study a natural asynchronous stochastic gradient method for the solution of minimization problems of the form

$$\text{minimize } f(x) := \mathbb{E}_P[F(x; W)] = \int_\Omega F(x; \omega) dP(\omega), \tag{1}$$

where $x \mapsto F(x; \omega)$ is convex for each $\omega \in \Omega$, $P$ is a probability distribution on $\Omega$, and the vector $x \in \mathbb{R}^d$. Stochastic gradient techniques for the solution of problem (1) have a long history in optimization, starting from the early work of Robbins and Monro [19] and continuing on through Ermoliev [7], Polyak and Juditsky [16], and Nemirovski et al. [14]. The latter two show how certain long stepsizes and averaging techniques yield more robust and asymptotically optimal optimization schemes, and we show how their results extend to practical parallel and asynchronous settings.

We consider an extension of previous stochastic gradient methods to a natural family of *asynchronous* gradient methods [3], where multiple processors can draw samples from the distribution $P$ and asynchronously perform updates to a centralized (shared) decision vector $x$. Our iterative scheme is based on the HOGWILD! algorithm of Niu et al. [15], which is designed to asynchronously solve certain stochastic optimization problems in multi-core environments, though our analysis and iterations are different. In particular, we study the following procedure, where each processor runs asynchronously and independently of the others, though they maintain a shared integer iteration counter $k$; each processor P asynchronously performs the following:

(i) Processor P reads current problem data $x$

(ii) Processor P draws a random sample $W \sim P$, computes $g = \nabla F(x; W)$, and increments the centralized counter $k$

(iii) Processor P updates $x \leftarrow x - \alpha_k g$ sequentially for each coordinate $j = 1, 2, \ldots, d$ by incrementing $[x]_j \leftarrow [x]_j - \alpha_k[g]_j$, where the scalars $\alpha_k$ are a non-increasing stepsize sequence.

Our main results show that because of the noise inherent to the sampling process for $W$, the errors introduced by asynchrony in iterations (i)–(iii) are asymptotically negligible: they do not matter. Even more, we can efficiently construct an $x$ from the asynchronous process possessing optimal convergence rate and asymptotic variance. This has consequences for solving stochastic optimization problems on multi-core and multi-processor systems; we can leverage parallel computing without performing any synchronization, so that given a machine with $m$ processors, we can read data and perform updates $m$ times as quickly as with a single processor, and the error from reading stale information on $x$ becomes asymptotically negligible. In Section 2, we state our main convergence theorems about the asynchronous iteration (i)–(iii) for solving problem (1). Our main result, Theorem 1, gives explicit conditions under which our results hold, and we give applications to specific stochastic optimization problems as well as a general result for asynchronous solution of operator equations. Roughly, all we require for our (optimal) convergence results is that the Hessian of $f$ be positive definite near $x^\star = \operatorname{argmin}_x f(x)$ and that the gradients $\nabla f(x)$ be smooth.

Several researchers have provided and analyzed asynchronous algorithms for optimization. Bertsekas and Tsitsiklis [3] provide a comprehensive study both of models of asynchronous computation and analyses of asynchronous numerical algorithms. More recent work has studied asynchronous gradient procedures, though it often imposes strong conditions on gradient sparsity, conditioning of the Hessian of $f$, or allowable types of asynchrony; as we show, none are essential. Niu et al. [15] propose HOGWILD! and show that under sparsity and smoothness assumptions (essentially, that the gradients $\nabla F(x; W)$ have a vanishing fraction of non-zero entries, that $f$ is strongly convex, and $\nabla F(x; \omega)$ is Lipschitz for all $\omega$), convergence guarantees similar to the synchronous case are possible; Agarwal and Duchi [1] showed under restrictive ordering assumptions that some delayed gradient calculations have negligible asymptotic effect; and Duchi et al. [4] extended Niu et al.'s results to a dual averaging algorithm that works for non-smooth, non strongly-convex problems, so long as certain gradient sparsity assumptions hold. Researchers have also investigated parallel coordinate descent solvers; Richtárik and Takáč [18] and Liu et al. [13] show how certain "near-separability" properties of an objective function $f$ govern convergence rate of parallel coordinate descent methods, the latter focusing on asynchronous schemes. As we show, large-scale stochastic optimization renders many of these problem assumptions unnecessary.

In addition to theoretical results, in Section 3 we give empirical results on the power of parallelism and asynchrony in the implementation of stochastic approximation procedures. Our experiments demonstrate two results: first, even in non-asymptotic finite-sample settings, asynchrony introduces little degradation in solution quality, regardless of data sparsity (a common assumption in previous analyses); that is, asynchronously-constructed estimates are statistically efficient. Second, we show that there is some subtlety in implementation of these procedures in real hardware; while increases in parallelism lead to concomitant linear improvements in the speed with which we compute solutions to problem (1), in some cases we require strategies to reduce hardware resource competition between processors to achieve the full benefits of asynchrony.

**Notation**   A sequence of random variables or vectors $X_n$ converges in distribution to $Z$, denoted $X_n \xrightarrow{d} Z$, if $\mathbb{E}[f(X_n)] \to \mathbb{E}[f(Z)]$ for all bounded continuous functions $f$. We let $X_n \xrightarrow{p} Z$ denote convergence in probability, meaning that $\lim_n \mathbb{P}(\|X_n - Z\| > \epsilon) = 0$ for any $\epsilon > 0$. The notation $\mathsf{N}(\mu, \Sigma)$ denotes the multivariate Gaussian with mean $\mu$ and covariance $\Sigma$.

## 2   Main results

Our main results repose on a few standard assumptions often used for the analysis of stochastic optimization procedures, which we now detail, along with a few necessary definitions. We let $k$ denote the iteration counter used throughout the asynchronous gradient procedure. Given that we compute $g = \nabla F(x; W)$ with counter value $k$ in the iterations (i)–(iii), we let $x_k$ denote *the* (possibly inconsistent) particular $x$ used to compute $g$, and likewise say that $g = g_k$, noting that the update to $x$ is then performed using $\alpha_k$. In addition, throughout paper, we assume there is some *finite* bound $M < \infty$ such that no processor reads information more than $M$ steps out of date.

### 2.1   Asynchronous convex optimization

We now present our main theoretical results for solving the stochastic convex problem (1), giving the necessary assumptions on $f$ and $F(\cdot; W)$ for our results. Our first assumption roughly states that $f$ has quadratic expansion near the (unique) optimal point $x^\star$ and is smooth.

**Assumption A.** *The function $f$ has unique minimizer $x^\star$ and is twice continuously differentiable in the neighborhood of $x^\star$ with positive definite Hessian $H = \nabla^2 f(x^\star) \succ 0$ and there is a covariance matrix $\Sigma \succ 0$ such that*

$$\mathbb{E}[\nabla F(x^\star; W) \nabla F(x^\star; W)^\top] = \Sigma.$$

*Additionally, there exists a constant $C < \infty$ such that the gradients $\nabla F(x; W)$ satisfy*

$$\mathbb{E}[\|\nabla F(x; W) - \nabla F(x^\star; W)\|^2] \leq C \|x - x^\star\|^2 \quad \text{for all } x \in \mathbb{R}^d. \tag{2}$$

*Lastly, $f$ has $L$-Lipschitz continuous gradient: $\|\nabla f(x) - \nabla f(y)\| \leq L \|x - y\|$ for all $x, y \in \mathbb{R}^d$.*

Assumption A guarantees the uniqueness of the vector $x^\star$ minimizing $f(x)$ over $\mathbb{R}^d$ and ensures that $f$ is well-behaved enough for our asynchronous iteration procedure to introduce negligible noise over a non-asynchronous procedure. In addition to Assumption A, we make one of two additional assumptions. In the first case, we assume that $f$ is strongly convex:

**Assumption B.** *The function $f$ is $\lambda$-strongly convex over all of $\mathbb{R}^d$ for some $\lambda > 0$, that is,*

$$f(y) \geq f(x) + \langle \nabla f(x), y - x \rangle + \frac{\lambda}{2} \|x - y\|^2 \quad \text{for } x, y \in \mathbb{R}^d. \tag{3}$$

Our alternate assumption is a Lipschitz assumption on $f$ itself, made by virtue of a second moment bound on $\nabla F(x; W)$.

**Assumption B'.** *There exists a constant $G < \infty$ such that for all $x \in \mathbb{R}^d$,*

$$\mathbb{E}[\|\nabla F(x; W)\|^2] \leq G^2. \tag{4}$$

With our assumptions in place, we state our main theorem.

**Theorem 1.** *Let the iterates $x_k$ be generated by the asynchronous process* (i), (ii), (iii) *with stepsize choice $\alpha_k = \alpha k^{-\beta}$, where $\beta \in (\frac{1}{2}, 1)$ and $\alpha > 0$. Let Assumption A and either of Assumptions B or B' hold. Then*

$$\frac{1}{\sqrt{n}} \sum_{k=1}^{n} (x_k - x^\star) \overset{d}{\to} \mathsf{N}\left(0, H^{-1} \Sigma H^{-1}\right) = \mathsf{N}\left(0, (\nabla^2 f(x^\star))^{-1} \Sigma (\nabla^2 f(x^\star))^{-1}\right).$$

Before moving to example applications of Theorem 1, we note that its convergence guarantee is generally unimprovable even by numerical constants. Indeed, for classical statistical problems, the covariance $H^{-1} \Sigma H^{-1}$ is the inverse Fisher information, and by the Le Cam-Hájek local minimax theorems [9] and results on Bahadur efficiency [21, Chapter 8], this is the optimal covariance matrix, and the best possible rate is $n^{-\frac{1}{2}}$. As for function values, using the delta method [e.g. 10, Theorem 1.8.12], we can show the optimal convergence rate of $1/n$ on function values.

**Corollary 1.** *Let the conditions of Theorem 1 hold. Then $n\left(f\left(\frac{1}{n}\sum_{k=1}^{n} x_k\right) - f(x^\star)\right) \overset{d}{\to} \frac{1}{2} \operatorname{tr}\left[H^{-1}\Sigma\right] \cdot \chi_1^2$, where $\chi_1^2$ denotes a chi-squared random variable with $1$ degree of freedom, and $H = \nabla^2 f(x^\star)$ and $\Sigma = \mathbb{E}[\nabla F(x^\star; W) \nabla F(x^\star; W)^\top]$.*

## 2.2 Examples

We now give two classical statistical optimization problems to illustrate Theorem 1. We verify that the conditions of Assumptions A and B or B' are not overly restrictive.

**Linear regression** Standard linear regression problems satisfies the conditions of Assumption B. In this case, the data $\omega = (a, b) \in \mathbb{R}^d \times \mathbb{R}$ and the objective $F(x; \omega) = \frac{1}{2}(\langle a, x \rangle - b)^2$. If we have moment bounds $\mathbb{E}[\|a\|_2^4] < \infty$, $\mathbb{E}[b^2] < \infty$ and $H = \mathbb{E}[aa^\top] \succ 0$, we have $\nabla^2 f(x^\star) = H$, and the assumptions of Theorem 1 are certainly satisfied. Standard modeling assumptions yield more concrete guarantees. For example, if $b = \langle a, x^\star \rangle + \varepsilon$ where $\varepsilon$ is independent mean-zero noise with $\mathbb{E}[\varepsilon^2] = \sigma^2$, the minimizer of $f(x) = \mathbb{E}[F(x; W)]$ is $x^\star$, we have $\langle a, x^\star \rangle - b = -\varepsilon$, and

$$\mathbb{E}[\nabla F(x^\star; W) \nabla F(x^\star; W)^\top] = \mathbb{E}[(\langle a, x^\star \rangle - b) aa^\top (\langle a, x^\star \rangle - b)] = \mathbb{E}[aa^\top \varepsilon^2] = \sigma^2 \mathbb{E}[aa^\top] = \sigma^2 H.$$

In particular, the asynchronous iterates satisfy

$$\frac{1}{\sqrt{n}} \sum_{k=1}^{n} (x_k - x^\star) \overset{d}{\to} \mathsf{N}(0, \sigma^2 H^{-1}) = \mathsf{N}\left(0, \sigma^2 \mathbb{E}[aa^\top]^{-1}\right),$$

which is the (minimax optimal) asymptotic covariance of the ordinary least squares estimate of $x^\star$.

**Logistic regression** As long as the data has finite second moment, logistic regression problems satisfy all the conditions of Assumption B' in Theorem 1. We have $\omega = (a, b) \in \mathbb{R}^d \times \{-1, 1\}$ and instantaneous objective $F(x; \omega) = \log(1 + \exp(-b \langle a, x \rangle))$. For fixed $\omega$, this function is Lipschitz continuous and has gradient and Hessian

$$\nabla F(x; \omega) = -\frac{1}{1 + \exp(b \langle a, x \rangle)} ba \quad \text{and} \quad \nabla^2 F(x; \omega) = \frac{e^{b \langle a, x \rangle}}{(1 + e^{b \langle a, x \rangle})^2} aa^\top,$$

where $\nabla F(x; \omega)$ is Lipschitz continuous as $\|\nabla^2 F(x; (a, b))\| \leq \frac{1}{4} \|a\|_2^2$. So long as $\mathbb{E}[\|a\|_2^2] < \infty$ and $\mathbb{E}[\nabla^2 F(x^\star; W)] \succ 0$ (i.e. $\mathbb{E}[aa^\top]$ is positive definite), Theorem 1 applies to logistic regression.

## 2.3 Extension to nonlinear problems

We prove Theorem 1 by way of a more general result on finding the zeros of a residual operator $R : \mathbb{R}^d \to \mathbb{R}^d$, where we only observe noisy views of $R(x)$, and there is unique $x^\star$ such that $R(x^\star) = 0$. Such situations arise, for example, in the solution of stochastic monotone operator problems (cf. Juditsky, Nemirovski, and Tauvel [8]). In this more general setting, we consider the following asynchronous iterative process, which extends that for the convex case outlined previously. Each processor P performs the following asynchronously and independently:

(i) Processor P reads current problem data $x$

(ii) Processor P receives vector $g = R(x) + \xi$, where $\xi$ is a random (conditionally) mean-zero noise vector, and increments a centralized counter $k$

(iii) Processor P updates $x \leftarrow x - \alpha_k g$ sequentially for each coordinate $j = 1, 2, \ldots, d$ by incrementing $[x]_j = [x]_j - \alpha_k [g]_j$.

As in the convex case, we associate vectors $x_k$ and $g_k$ with the update performed using $\alpha_k$, and we let $\xi_k$ denote the noise vector used to construct $g_k$. These iterates and assignment of indices imply that $x_k$ has the form

$$x_k = -\sum_{i=1}^{k-1} \alpha_i E^{ki} g_i, \tag{5}$$

where $E^{ki} \in \{0, 1\}^{d \times d}$ is a diagonal matrix whose $j$th diagonal entry captures that coordinate $j$ of the $i$th gradient has been incorporated into iterate $x_k$.

We define the an increasing sequence of $\sigma$-fields $\mathcal{F}_k$ by

$$\mathcal{F}_k = \sigma \left( \xi_1, \ldots, \xi_k, \{ E^{ij} : i \leq k + 1, j \leq i \} \right), \tag{6}$$

that is, the noise variables $\xi_k$ are adapted to the filtration $\mathcal{F}_k$, and these $\sigma$-fields are the smallest containing both the noise and all index updates that have occurred and that will occur to compute $x_{k+1}$. Thus we have $x_{k+1} \in \mathcal{F}_k$, and our mean-zero assumption on the noise $\xi$ is

$$\mathbb{E}[\xi_k \mid \mathcal{F}_{k-1}] = 0.$$

We base our analysis on Polyak and Juditsky's study [16] of stochastic approximation procedures, so we enumerate a few more requirements—modeled on theirs—for our results on convergence of the asynchronous iterations for solving the nonlinear equality $R(x^\star) = 0$. We assume there is a Lyapunov function $V$ satisfying $V(x) \geq \lambda \|x\|^2$ for all $x \in \mathbb{R}^d$, $\|\nabla V(x) - \nabla V(y)\| \leq L \|x - y\|$ for all $x, y$, that $\nabla V(0) = 0$, and $V(0) = 0$. This implies

$$\lambda \|x\|^2 \leq V(x) \leq V(0) + \langle \nabla V(0), x - 0 \rangle + \frac{L}{2} \|x\|^2 = \frac{L}{2} \|x\|^2 \tag{7}$$

and $\|\nabla V(x)\|^2 \leq L^2 \|x\|^2 \leq (L^2/\lambda) V(x)$. We make the following assumptions on the residual $R$.

**Assumption C.** *There exists a matrix $H \in \mathbb{R}^{d \times d}$ with $H \succ 0$, a parameter $0 < \gamma \leq 1$, constant $C < \infty$, and $\epsilon > 0$ such that if $x$ satisfies $\|x - x^\star\| \leq \epsilon$,*

$$\|R(x) - H(x - x^\star)\| \leq C \|x - x^\star\|^{1+\gamma}.$$

Assumption C essentially requires that $R$ is differentiable at $x^\star$ with derivative matrix $H \succ 0$. We also make a few assumptions on the noise process $\xi$; specifically, we assume $\xi$ implicitly depends on $x \in \mathbb{R}^d$ (so that we may write $\xi_k = \xi(x_k)$), and that the following assumption holds.

**Assumption D.** *The noise vector $\xi(x)$ decomposes as $\xi(x) = \xi(0) + \zeta(x)$, where $\xi(0)$ is a process satisfying $\mathbb{E}[\xi_k(0)\xi_k(0)^\top \mid \mathcal{F}_{k-1}] \xrightarrow{p} \Sigma \succ 0$ for a matrix $\Sigma \in \mathbb{R}^{d \times d}$, $\sup_k \mathbb{E}[\|\xi_k(0)\|^2 \mid \mathcal{F}_{k-1}] < \infty$ with probability 1, and $\mathbb{E}[\|\zeta_k(x)\|^2 \mid \mathcal{F}_{k-1}] \leq C \|x - x^\star\|^2$ for a constant $C < \infty$ and all $x \in \mathbb{R}^d$.*

As in the convex case, we make one of two additional assumptions, which should be compared with Assumptions B and B'. The first is that $R$ gives globally strong information about $x^\star$.

**Assumption E** (Strongly convex residuals). *There exists a constant $\lambda_0 > 0$ such that for all $x \in \mathbb{R}^d$, $\langle \nabla V(x - x^\star), R(x) \rangle \geq \lambda_0 V(x - x^\star)$.*

Alternatively, we may make an assumption on the boundedness of $R$, which we shall see suffices for proving our main results.

**Assumption E'** (Bounded residuals). *There exist $\lambda_0 > 0$ and $\epsilon > 0$ such that*

$$\inf_{0 < \|x - x^\star\| \leq \epsilon} \frac{\langle \nabla V(x - x^\star), R(x) \rangle}{V(x - x^\star)} \geq \lambda_0 \quad and \quad \inf_{\epsilon < \|x - x^\star\|} \langle \nabla V(x - x^\star), R(x) \rangle > 0.$$

*In addition there exists $C < \infty$ such that, $\|R(x)\| \leq C$ and $\mathbb{E}[\|\xi_k\|^2 \mid \mathcal{F}_{k-1}] \leq C^2$ for all $k$ and $x$.*

With these assumptions in place, we obtain the following more general version of Theorem 1; indeed, we show that Theorem 1 is a consequence of this result.

**Theorem 2.** *Let $V$ be a function satisfying inequality* (7), *and let Assumptions C and D hold. Let the stepsizes $\alpha_k = \alpha k^{-\beta}$, where $\frac{1}{1+\gamma} < \beta < 1$. Let one of Assumptions E or E' hold. Then*

$$\frac{1}{\sqrt{n}} \sum_{k=1}^{n} (x_k - x^\star) \xrightarrow{d} \mathsf{N}\left(0, H^{-1}\Sigma H^{-1}\right).$$

We may compare this result to Polyak and Juditsky's Theorem 2 [16], which gives identical asymptotic convergence guarantees but with somewhat weaker conditions on the function $V$ and stepsize sequence $\alpha_k$. Our stronger assumptions, however, allow our result to apply even in fully asynchronous settings.

## 2.4 Proof sketch

We provide rigorous proofs in the long version of this paper [5], providing an amputated sketch here. First, to show that Theorem 1 follows from Theorem 2, we set $R(x) = \nabla f(x)$ and $V(x) = \frac{1}{2}\|x\|^2$. We can then show that Assumption A, which guarantees a second-order Taylor expansion, implies Assumption C with $\gamma = 1$ and $H = \nabla^2 f(x^\star)$. Moreover, Assumption B (or B') implies Assumption E (respectively, E'), while to see that Assumption D holds, we set $\xi(0) = \nabla F(x^\star; W)$, taking $\Sigma = \mathbb{E}[\nabla F(x^\star; W)\nabla F(x^\star; W)^\top]$ and $\zeta(x) = \nabla F(x; W) - \nabla F(x^\star; W)$, and applying inequality (2) of Assumption A to satisfy Assumption D with the vector $\zeta$.

The proof of Theorem 2 is somewhat more involved. Roughly, we show the asymptotic equivalence of the sequence $x_k$ from expression (5) to the easier to analyze sequence $\widetilde{x}_k = -\sum_{i=1}^{k-1} \alpha_i g_i$. Asymptotically, we obtain $\mathbb{E}[\|x_k - \widetilde{x}_k\|^2] = O(\alpha_k^2)$, while the iterates $\widetilde{x}_k$—in spite of their incorrect gradient calculations—are close enough to a correct stochastic gradient iterate that they possess optimal asymptotic normality properties. This "close enough" follows by virtue of the squared error bounds for $\zeta$ in Assumption D, which guarantee that $\xi_k$ essentially behaves like an i.i.d. sequence asymptotically (after application of the Robbins-Siegmund martingale convergence theorem [20]), which we then average to obtain a central-limit-theorem.

# 3 Experimental results

We provide empirical results studying the performance of asynchronous stochastic approximation schemes on several simulated and real-world datasets. Our theoretical results suggest that asynchrony should introduce little degradation in solution quality, which we would like to verify; we

also investigate the engineering techniques necessary to truly leverage the power of asynchronous stochastic procedures. In our experiments, we focus on linear and logistic regression, the examples given in Section 2.2; that is, we have data $(a_i, b_i) \in \mathbb{R}^d \times \mathbb{R}$ (for linear regression) or $(a_i, b_i) \in \mathbb{R}^d \times \{-1, 1\}$ (for logistic regression), for $i = 1, \ldots, N$, and objectives

$$f(x) = \frac{1}{2N} \sum_{i=1}^{N} (\langle a_i, x \rangle - b_i)^2 \quad \text{and} \quad f(x) = \frac{1}{N} \sum_{i=1}^{N} \log\left(1 + \exp(-b_i \langle a_i, x \rangle)\right). \tag{8}$$

We perform each of our experiments using a 48-core Intel Xeon machine with 1 terabyte of RAM, and have put code and binaries to replicate our experiments on CodaLab [6]. The Xeon architecture puts each core onto one of four sockets, where each socket has its own memory. To limit the impact of communication overhead in our experiments, we limit all experiments to at most 12 cores, all on the same socket. Within an experiment—based on the empirical expectations (8)—we iterate in *epochs*, meaning that our stochastic gradient procedure repeatedly loops through all examples, each exactly once.[1] Within an epoch, we use a fixed stepsize $\alpha$, decreasing the stepsize by a factor of .9 between each epoch (this matches the experimental protocol of Niu et al. [15]). Within each epoch, we choose examples in a randomly permuted order, where the order changes from epoch to epoch (cf. [17]). To address issues of hardware resource contention (see Section 3.2 for more on this), in some cases we use a *mini-batching* strategy. Abstractly, in the formulation of the basic problem (1), this means that in each calculation of a stochastic gradient $g$ we draw $\mathsf{B} \geq 1$ samples $W_1, \ldots, W_\mathsf{B}$ i.i.d. according to $P$, then set

$$g(x) = \frac{1}{\mathsf{B}} \sum_{b=1}^{\mathsf{B}} \nabla F(x; W_b). \tag{9}$$

The mini-batching strategy (9) does not change the (asymptotic) convergence guarantees of asynchronous stochastic gradient descent, as the covariance matrix $\Sigma = \mathbb{E}[g(x^\star)g(x^\star)^\top]$ satisfies $\Sigma = \frac{1}{\mathsf{B}} \mathbb{E}[\nabla F(x^\star; W) \nabla F(x^\star; W)^\top]$, while the total iteration count is reduced by the a factor $\mathsf{B}$. Lastly, we measure the performance of optimization schemes via *speedup*, defined as

$$\text{speedup} = \frac{\text{average epoch runtime on a single core using HOGWILD!}}{\text{average epoch runtime on } m \text{ cores}}. \tag{10}$$

In our experiments, as increasing the number $m$ of cores does not change the gap in optimality $f(x_k) - f(x^\star)$ after each epoch, speedup is equivalent to the ratio of the time required to obtain an $\epsilon$-accurate solution using a single processor/core to that required to obtain $\epsilon$-accurate solution using $m$ processors/cores.

## 3.1 Efficiency and sparsity

For our first set of experiments, we study the effect that data sparsity has on the convergence behavior of asynchronous methods—sparsity has been an essential part of the analysis of many asynchronous and parallel optimization schemes [15, 4, 18], while our theoretical results suggest it should be unimportant—using the linear regression objective (8). We generate synthetic linear regression problems with $N = 10^6$ examples in $d = 10^3$ dimensions via the following procedure. Let $\rho \in (0, 1]$ be the desired fraction of non-zero gradient entries, and let $\Pi_\rho$ be a random projection operator that zeros out all but a fraction $\rho$ of the elements of its argument, meaning that for $a \in \mathbb{R}^d$, $\Pi_\rho(a)$ uniformly at random chooses $\rho d$ elements of $a$, leaves them identical, and zeroes the remaining elements. We generate data for our linear regression drawing a random vector $u^\star \sim \mathsf{N}(0, I)$, then constructing $b_i = \langle a_i, u^\star \rangle + \varepsilon_i$, $i = 1, \ldots, N$, where $\varepsilon_i \overset{\text{i.i.d.}}{\sim} \mathsf{N}(0, 1)$, $a_i = \Pi_\rho(\widetilde{a}_i)$, $\widetilde{a}_i \overset{\text{i.i.d.}}{\sim} \mathsf{N}(0, I)$, and $\Pi_\rho(\widetilde{a}_i)$ denotes an independent random sparse projection of $\widetilde{a}_i$. To measure optimality gap, we directly compute $x^\star = (A^T A)^{-1} A^T b$, where $A = [a_1 \ a_2 \ \cdots \ a_N]^\top \in \mathbb{R}^{N \times d}$.

In Figure 1, we plot the results of simulations using densities $\rho \in \{.005, .01, .2, 1\}$ and mini-batch size $\mathsf{B} = 10$, showing the gap $f(x_k) - f(x^\star)$ as a function of the number of epochs for each of the given sparsity levels. We give results using 1, 2, 4, and 10 processor cores (increasing degrees of asynchrony), and from the plots, we see that regardless of the number of cores, the convergence

behavior is nearly identical, with very minor degradations in performance for the sparsest data. (We plot the gaps $f(x_k) - f(x^\star)$ on a logarithmic axis.) Moreover, as the data becomes denser, the more asynchronous methods—larger number of cores—achieve performance essentially identical to the fully synchronous method in terms of convergence versus number of epochs. In Figure 2, we plot the speedup achieved using different numbers of cores. We also include speedup achieved using multiple cores with explicit synchronization (locking) of the updates, meaning that instead of allowing asynchronous updates, each of the cores globally locks the decision vector when it reads, unlocks and performs mini-batched gradient computations, and locks the vector again when it updates the vector. We can see that the performance curve is much worse than than the without-locking performance curve across all densities. That the locking strategy also gains some speedup when the density is higher is likely due to longer computation of the gradients. However, the locking-strategy performance is still not competitive with that of the without-locking strategy.

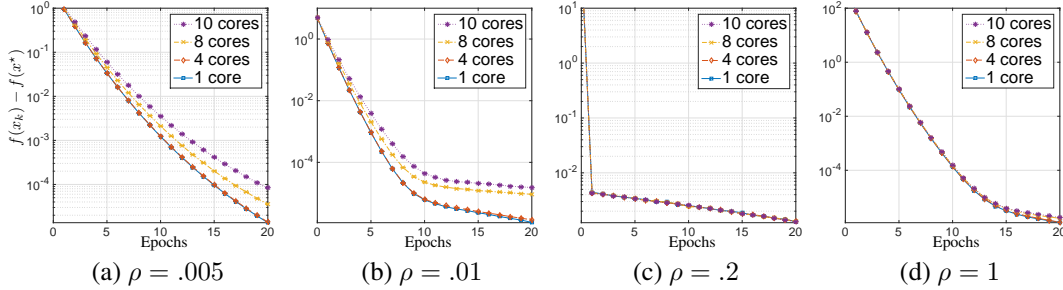

(a) $\rho = .005$     (b) $\rho = .01$     (c) $\rho = .2$     (d) $\rho = 1$

**Figure 1.** (Exponential backoff stepsizes) Optimality gaps for synthetic linear regression experiments showing effects of data sparsity and asynchrony on $f(x_k) - f(x^\star)$. A fraction $\rho$ of each vector $a_i \in \mathbb{R}^d$ is non-zero.

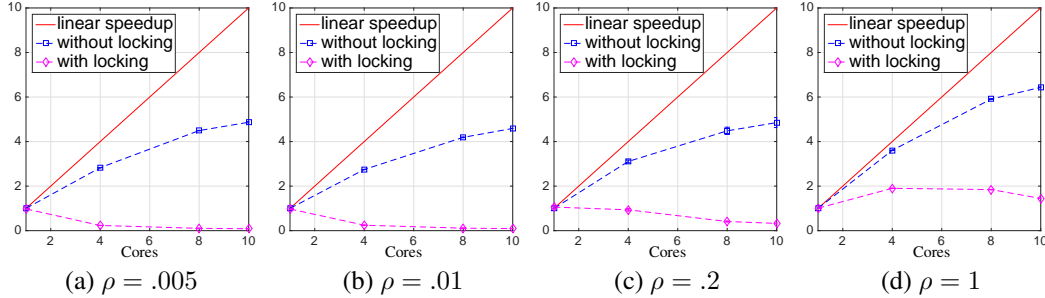

(a) $\rho = .005$     (b) $\rho = .01$     (c) $\rho = .2$     (d) $\rho = 1$

**Figure 2.** (Exponential backoff stepsizes) Speedups for synthetic linear regression experiments showing effects of data sparsity on speedup (10). A fraction $\rho$ of each vector $a_i \in \mathbb{R}^d$ is non-zero.

## 3.2 Hardware issues and cache locality

We detail a small set of experiments investigating hardware issues that arise even in implementation of asynchronous gradient methods. The Intel x86 architecture (as with essentially every processor architecture) organizes memory in a hierarchy, going from L1 to L3 (level 1 to level 3) caches of increasing sizes. An important aspect of the speed of different optimization schemes is the relative fraction of memory *hits*, meaning accesses to memory that is cached locally (in order of decreasing speed, L1, L2, or L3 cache). In Table 1, we show the proportion of cache misses at each level of the memory hierarchy for our synthetic regression experiment with fully dense data ($\rho = 1$) over the execution of 20 epochs, averaged over 10 different experiments. We compare memory contention when the batch size B used to compute the local asynchronous gradients (9) is 1 and 10. We see that the proportion of misses for the fastest two levels—1 and 2—of the cache for B = 1 increase significantly with the number of cores, while increasing the batch size to B = 10 substantially mitigates cache incoherency. In particular, we maintain (near) linear increases in iteration speed with little degradation in solution quality (the gap $f(\widehat{x}) - f(x^\star)$ output by each of the procedures with and without batching is identical to within $10^{-3}$; cf. Figure 1(d)).

| | Number of cores | 1 | 4 | 8 | 10 |
|---|---|---|---|---|---|
| | fraction of L1 misses | 0.0009 | 0.0017 | 0.0025 | 0.0026 |
| | fraction of L2 misses | 0.5638 | 0.6594 | 0.7551 | 0.7762 |
| No batching (B = 1) | fraction of L3 misses | 0.6152 | 0.4528 | 0.3068 | 0.2841 |
| | epoch average time (s) | 4.2101 | 1.6577 | 1.4052 | 1.3183 |
| | **speedup** | **1.00** | **2.54** | **3.00** | **3.19** |
| | Number of cores | 1 | 4 | 8 | 10 |
| | fraction of L1 misses | 0.0012 | 0.0011 | 0.0011 | 0.0011 |
| | fraction of L2 misses | 0.5420 | 0.5467 | 0.5537 | 0.5621 |
| Batch size B = 10 | fraction of L3 misses | 0.5677 | 0.5895 | 0.5714 | 0.5578 |
| | epoch average time (s) | 4.4286 | 1.1868 | 0.6971 | 0.6220 |
| | **speedup** | **1.00** | **3.73** | **6.35** | **7.12** |

**Table 1.** Memory traffic for batched updates (9) versus non-batched updates (B = 1) for a dense linear regression problem in $d = 10^3$ dimensions with a sample of size $N = 10^6$. Cache misses are substantially higher with B = 1.

### 3.3 Real datasets

We perform experiments using three different real-world datasets: the Reuters RCV1 corpus [11], the Higgs detection dataset [2], and the Forest Cover dataset [12]. Each represents a binary classification problem which we formulate using logistic regression. We briefly detail statistics for each:

(1) Reuters RCV1 dataset consists of $N \approx 7.81 \cdot 10^5$ data vectors (documents) $a_i \in \{0,1\}^d$ with $d \approx 5 \cdot 10^4$ dimensions; each vector has sparsity approximately $\rho = 3 \cdot 10^{-3}$. Our task is to classify each document as being about corporate industrial topics (CCAT) or not.

(2) The Higgs detection dataset consists of $N = 10^6$ data vectors $\widetilde{a}_i \in \mathbb{R}^{d_0}$, with $d_0 = 28$. We quantize each coordinate into 5 bins containing equal fraction of the coordinate values and encode each vector $\widetilde{a}_i$ as a vector $a_i \in \{0,1\}^{5d_0}$ whose non-zero entries correspond to quantiles into which coordinates fall. The task is to detect (simulated) emissions from a linear accelerator.

(3) The Forest Cover dataset consists of $N \approx 5.7 \cdot 10^5$ data vectors $a_i \in \{-1,1\}^d$ with $d = 54$, and the task is to predict forest growth types.

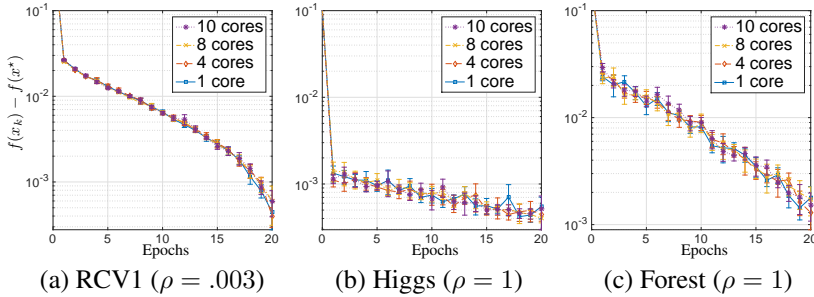

(a) RCV1 ($\rho = .003$)  (b) Higgs ($\rho = 1$)  (c) Forest ($\rho = 1$)

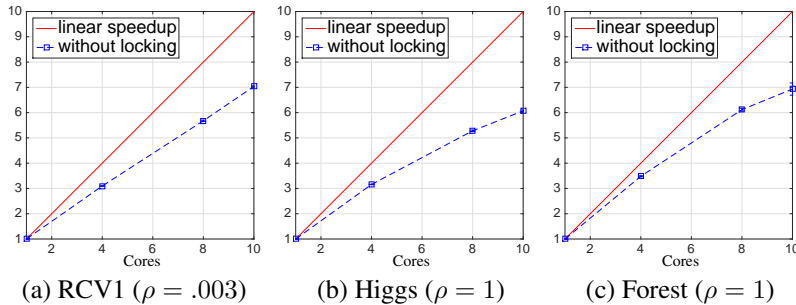

(a) RCV1 ($\rho = .003$)  (b) Higgs ($\rho = 1$)  (c) Forest ($\rho = 1$)

**Figure 3.** (Exponential backoff stepsizes) Optimality gaps $f(x_k) - f(x^\star)$ on the (a) RCV1, (b) Higgs, and (c) Forest Cover datasets.

**Figure 4.** (Exponential backoff stepsizes) Logistic regression experiments showing speedup (10) on the (a) RCV1, (b) Higgs, and (c) Forest Cover datasets.

In Figure 3, we plot the gap $f(x_k) - f(x^\star)$ as a function of epochs, giving standard error intervals over 10 runs for each experiment. There is essentially no degradation in objective value for the different numbers of processors, and in Figure 4, we plot speedup achieved using 1, 4, 8, and 10 cores with batch sizes B = 10. Asynchronous gradient methods achieve speedup of between $6\times$ and $8\times$ on each of the datasets using 10 cores.

## Footnotes

[1]Strictly speaking, this violates the stochastic gradient assumption, but it allows direct comparison with the original HOGWILD! code and implementation [15].

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
