[Reviews · NeurIPS 2015]

Submitted by Assigned_Reviewer_1

The author claim convergence rate O(1/sqrt(T)), without assumptions on the number of asynchronously working processors and with any amount of "sparsity" (partial separability). Considering that the subsequent empirical section shows that the convergence very much depends on the amount of sparsity, I would suggest the authors study the dependence of the sparsity and the number of processors as well, e.g. similar to http://arxiv.org/abs/1311.1873, http://arxiv.org/abs/1406.0238.
Summary: The study of asynchronous parallel algorithms is very challenging. The authors do some heavy lifting to obtain a middling result, but that should still be applauded.

Submitted by Assigned_Reviewer_2

This paper provides some theoretical analysis to show that completely asynchronous stochastic gradient procedures asymptotically achieve optimal convergence rates for the solution of convex optimization problems under nearly the same conditions required for asymptotic optimality of standard stochastic gradient procedures. Furthermore, empirical results are also demonstrated to verify the theoretical results.

The theoretical results of this paper seem to be promising and important. Furthermore, the experimental results successfully verify the strong performance of the asynchronous parallel stochastic optimization schemes, which is consistent with the theoretical analysis.

The results can be improved if faster convergence rates can also be provided with more strict assumptions, such as smooth and strongly convex.
Summary: The theoretical results of this paper seem to be promising, and the empirical results are consistent with the theoretical analysis.

Submitted by Assigned_Reviewer_3

The paper presents an asynchronous stochastic approximation algorithm, with in mind a distributed implementation. The proposed algorithm is described at the bottom of the first page, and this description is rather unclear. I think the authors should exactly describe how the various processors access the central memory, in both writing and reading. In particular I did not get how the centralized counter k was updated. Is k read along with the data x in step (i)? Please clarify.

This work extends the results of [1] to remove some of the technical assumptions made to ensure convergence. The authors need to elaborate further on the differences between the existing work, and in particular [1], and their work. For example, assumptions made should be discussed in more details. Right now the related work section is sketchy (12 lines on the second page).

The role of the centralized counter should also be emphasized. Was it present in the algorithms proposed in [1]? My guess is that having such counter does not matter at all, see for example http://arxiv.org/pdf/1505.04824.pdf and references therein.
Summary: The paper proposes an asynchronous stochastic approximation algorithm, and establishes its convergence rate. The description of the algorithm is not so clear to me, and its connection to existing algorithms and convergence results could be also clarified. The topic is interesting and important, and establishing convergence of such schemes under a minimal set of assumptions is important.

Submitted by Assigned_Reviewer_4

Summary of paper:

The paper studies a class of asynchronous methods to solving optimization problems in which the objective function is an integral functional of a random process or variable integrated against a probability measure. The class of problem assumed is that the argument of the functional for each non-zero measure set or outcome in the sample space upon which the measure is defined will produce a convex function. In this case the authors demonstrate that asymptotically, completely asynchronous stochastic gradient procedures achieve optimal convergence rates for the solution of convex optimization problems under nearly the same conditions required for asymptotic optimality of standard stochastic gradient procedures. The authors are able to then propose based on these results specialised asynchronous procedures based on multi-core parallelization schemes for stochastic gradient based optimization algorithms which behave analogously to synchronous procedures.

Quality:

The problem statement and all assumptions are very carefully set out and clearly written and explained with regard to the meaning and significance of such assumptions. The exposition and explanation of Theorem 1 is very well presented as is the technical details for Theorem 2 and the required Lemmas in the appendix.

The examples selected to illustrate the method were also very interesting having selected the regression examples based on linear regression as a toy illustration and then non-linear logistic regressions were proposed. These were relatively simple and demonstrated that the required conditions were not overly restrictive for real problems in applications settings.

However, it would be perhaps informative to explain a little more background on the context of the extended problems based on stochastic monotone operator problems, this is left very brief and a citation given, it would be useful to see a little more context here.

In assumption C and D for the distribution of the noise process it would be good to get more explanation of these assumptions and their significance and interpretation.

The experiment performed is very carefully detailed and well designed.

Clarity:

The paper is highly technical for a NIPS audience but it is very well written and explained carefully so it is still in my opinion in the scope of the conference. I believe the authors have done an excellent job to explain such detailed results in a short paper so clearly and concisely.

Originality:

The paper is based on several existing works but provides a useful set of results for practical problems which generalise and relax assumptions of existing works.

Significance:

The authors show that their results significantly generalize existing works that have considered similar results, though as they state previous works of a related nature often impose strong conditions on gradient sparsity or conditioning of the Hessian or allowable types of asynchrony. In particular they only require minimal conditions which are that at the argument that minimizes the objective function one has a positive definite evaluation of the function and that the gradient are smooth.

Summary: The paper is very well written and motivation for the problem statement and significance of the results derived is clearly explained. The results in Theorem 1 and Theorem 2 are clearly placed in context of their generalizations of existing works and then the example to illustrate the theoretical results in linear and logistic regressions is well designed and explained clearly.

Author Feedback
Author rebuttal: We thank each of the reviewers for their careful reading of the paper,
constructive feedback, and references provided. We address each reviewer's
responses in turn below, asking forgiveness for necessary brevity.

R1

Thanks for the comments and references to other work. Let us briefly discuss
the reviewer's comment that "convergence depends on sparsity" in the empirical
section. Indeed, the convergence behavior of SGD itself is different for
different amounts of sparsity--this is clear both theoretically and
experimentally, as noted by the reviewer. We show, however, that asynchrony
does not introduce any additional slowdown in convergence rate (as a function
of iteration) over a synchronous/one-thread method, so that there is
*relatively* no penalty. In Figures 1 and 3, the convergence (in number of
iterations) for each of the number of cores are nearly identical, which we
believe highlights that asynchrony does not hurt. And Figures 2 and 4 show
that it can, in fact, be very helpful.

R2

We thank the reviewer for the positive feedback.

We note in passing that our convergence guarantees are unimprovable and
optimal, even to numerical constant factors. (See the discussion after Theorem
1.) It will certainly be interesting to understand exactly what conditions are
necessary for variants and improvements of these rates of convergence.

R4

We appreciate the reviewer's enthusiasm and quite positive feedback--we
tried hard to make the paper approachable given the length constraints.

The reviewer's remark that our elaboration on monotone operator problems is
sparse is well taken--we will give more in our revision. As a short tidbit
along these lines, we note Bertsekas' and Tsitsiklis's book [reference 3,
Chapter 3.5]; a few natural problems include stochastic games and saddle-point
problems, systems of equations, and (stochastic) network equilibrium problems.

R6

Thanks for the quick feedback.

R7

We thank the reviewer for the review and feedback, and hope to clarify a few
of the issues the reviewer raises. Our discussion of related work was brief
due to the constraints of the 8 page conference paper. In short, previous work
on asynchronous algorithms generally (1) makes somewhat strong assumptions on
problem structure such as gradient sparsity, variants of diagonal dominance of
the Hessian matrix, or ordering of updates (refs. [1,3,4,11,13,16]), or (2)
left open interesting questions of the possibilities of optimal convergence
rates (e.g. ref. [3]). We attempt to address both of these issues. (The paper
[1] in particular assumes that the updates are performed in fixed
round-robin-like orders, and also does not give as sharp of optimality
guarantees as we do.)

In terms of memory access, we assume that our computer is a shared memory
system, where each core has access to the same memory, though we do not
believe this is an essential part of our contribution. The memory model of Niu
et al.'s Hogwild! is essentially identical to ours. Broadly, our contribution
is to show that asynchrony has asymptotically negligible effect on the
optimality guarantees of stochastic gradient procedures, and this is quite
robust to delay models.

The issue of the centralized counter is a good question; we will clarify that
this counter is a single integer shared between processors, accessed
asynchronously in the same way that the vector $x$ is read and written (so it
is read with the data in step (i)). Most modern processors allow an atomic
increment, which we assume on $k$. All stochastic gradient procedures we know
either assume a fixed number of updates--so that each processor knows a
stepsize in advance--or have decreasing stepsizes based on something like a
counter. Other work on asynchronous procedures makes similar assumptions. The
algorithms of [1] (see Eq. (9) of http://arxiv.org/pdf/1104.5525.pdf) have a
stepsize $\alpha(t)$ that decreases with a (known centralized) iteration $t$.
The paper http://arxiv.org/abs/1505.04824 assumes a bounded delay
$\tau_{\max}$ on pg. 8, and Alg. 1 has a stepsize that is implicitly global
because of this delay. We could change our analysis to have local counters
$k$, but this adds--in our view--unnecessary complexity.